# Development of Solvent-Free Co-Ground Method to Produce Terbinafine Hydrochloride Cyclodextrin Binary Systems; Structural and In Vitro Characterizations

**DOI:** 10.3390/pharmaceutics14040744

**Published:** 2022-03-30

**Authors:** Balázs Attila Kondoros, Orsolya Jójárt-Laczkovich, Ottó Berkesi, Piroska Szabó-Révész, Ildikó Csóka, Rita Ambrus, Zoltán Aigner

**Affiliations:** 1Institute of Pharmaceutical Technology and Regulatory Affairs, Faculty of Pharmacy, University of Szeged, H-6720 Szeged, Hungary; kondoros.balazs.attila@szte.hu (B.A.K.); jojartne.laczkovich.orsolya@szte.hu (O.J.-L.); reveszpiroska@szte.hu (P.S.-R.); csoka.ildiko@szte.hu (I.C.); aigner.zoltan@szte.hu (Z.A.); 2Department of Physical Chemistry and Materials Science, Faculty of Science and Informatics, University of Szeged, H-6720 Szeged, Hungary; oberkesi@chem.u-szeged.hu

**Keywords:** HPBCD, DIMEB, XRPD, DSC, Raman, mapping, FTIR, peak-fitting, dissolution studies, diffusion studies

## Abstract

Molecular complexation with cyclodextrins (CDs) has long been a known process for modifying the physicochemical properties of problematic active pharmaceutical ingredients with poor water solubility. In current times, the focus has been on the solvent-free co-grinding process, which is an industrially feasible process qualifying as a green technology. In this study, terbinafine hydrochloride (TER), a low solubility antifungal drug was used as a model drug. This study aimed to prepare co-ground products and follow through the preparation process of the co-grinding method in the case of TER and two amorphous CD derivatives: (2-hydroxypropyl)-β-cyclodextrin (HPBCD); heptakis-(2,6-di-O-methyl)-β-cyclodextrin (DIMEB). For this evaluation, the following analytical tools and methods were used: phase solubility studies, differential scanning calorimetry (DSC), X-ray powder diffractometry (XRPD), hot-stage X-ray powder diffractometry (HOT-XRPD), Fourier-transform infrared (FT-IR), Raman spectroscopy, and Scanning Electron Microscopy (SEM). Furthermore, in vitro characterization (dissolution and diffusion studies) was performed in two kinds of dissolution medium without enzymes. In the XRPD and SEM studies, it was found that the co-grinding of the components resulted in amorphous products. FT-IR and Raman spectroscopies confirmed the formation of an inclusion complex through the unsaturated aliphatic chain of TER and CDs. In vitro characterization suggested better dissolution properties for both CDs and decreased diffusion at higher pH levels in the case of HPBCD.

## 1. Introduction

Improving the physicochemical properties of poorly soluble drugs represents one of the most difficult challenges in the field of pharmaceutical technology [1]. Active pharmaceutical ingredients (APIs) are classified into four classes by the Biopharmaceutical Classification System (BCS) based on their solubility and permeability properties [2]. Drugs with poor water solubility are in Classes II and IV. Terbinafine is a lipophilic substance, and is slightly soluble in water, and according to the BCS, terbinafine and its hydrochloride salt, terbinafine hydrochloride (TER) belong to Class II [3]. The solubility of the active ingredient is strongly pH-dependent, as it is significantly less soluble at a higher pH [4]. TER is an allylamine antifungal agent which inhibits the squalene epoxidase enzyme [5]. This drug is suitable for both systemic and topical administration [6]. It accumulates in the hair and nails during systemic administration [7]. Various side effects may occur during therapy: gastrointestinal complaints, taste loss, and cutaneous adverse effects [8,9,10]. Elevation in liver enzymes and bilirubin have also been observed, and acute hepatitis has been reported [11].

CDs are widely used in several industrial sectors including cosmetic, food, and pharmaceutical industries [12]. CDs are a family of cyclic oligosaccharides and due to the chair conformation of the glucopyranoside units, CD bears a truncated cone or conical shape. In the wider end of the torus, the secondary hydroxyl functions of the sugar residues are located, while the primary hydroxyl functions are located around the narrower lower margin. The central cavity of the molecule is instead coated with skeletal carbons and ethereal oxygens of the glucose residue. As a result of this structure, host–guest interactions can be formed between CDs and a wide variety of hydrophobic guest molecules [13]. The reason for the popularity of CDs is that they are able to form an inclusion complex with many active ingredients and thus improve their physicochemical properties (solubility, stability, bioavailability, etc.) [14]. The mechanism for this complexation is based on non-covalent dynamic inclusion complex formation [15].

Conventional methods (kneading, co-evaporation, spray-drying, co-precipitation, etc.) use a certain amount of organic solvent [16]. However, removal of the residual solvent from these solid products often causes problems and additional expenses [17]. On the other hand, several recent studies have proved the applicability of the solvent-free co-grinding method in the case of numerous drugs and CDs [18,19,20,21]. Using this method, complexation is performed via mechanochemical action rather than using solvents [22]. The co-grinding of active substances and CDs often results in a reduction of the degree of crystallinity or an amorphous product [23,24]. The physicochemical properties of the materials used and the parameters set during grinding (time and intensity) all influence the crystallinity of the final product [22,23]. Generally, a longer grinding time and higher intensity are required to amorphize the products, but using crystalline CDs lead to only partial amorphization and drug complexation [18]. Several articles deal with the improvement of the physicochemical properties of TER, and among these in many cases this goal is achieved by cyclodextrin (CD) complexation [4,25]. However, in these cases, all of these complexes are prepared by methods that require the use of an organic solvent.

The novelty of our work was to investigate co-grinding under laboratory conditions using comprehensive analytical methods for fast-screening the process during preformulation. According to articles on co-grinding, a combination of several analytical methods is required to design a cyclodextrin product [16]. In this study TER was chosen as a model drug and as excipients, two amorphous CD-derivatives were used, (2-hydroxy)propyl-β-cyclodextrin and heptakis(2,6-di-O-methyl)-β-cyclodextrin (DIMEB). Different analytical procedures were used to describe the process, and by evaluating these data together, we inferred the changes during the process. Phase solubility studies, differential scanning calorimetry (DSC), X-ray powder diffractometry (XRPD), hot-stage X-ray powder diffractometry (HOT-XRPD) Fourier-transform infrared spectroscopy (FT-IR), and Raman spectroscopy were used to examine physicochemical properties. In vitro dissolution and diffusion studies were applied to investigate solubility properties and to model the passive diffusion of the API.

## 2. Materials and Methods

### 2.1. Materials

Terbinafine hydrochloride (TER): (E)-N,6,6-trimethyl-N-(naphthalene-1-ylmethyl) hept-2-en-4-yn-1-amine hydrochloride was kindly donated by Gedeon Richter Plc. (Budapest, Hungary). Heptakis-(2,6-di-O-methyl)-β-cyclodextrin (degree of substitution: 14.00; molecular weight: 1331.0 g mol^−1^) was obtained from Cyclolab R&D Laboratory Ltd. (Budapest, Hungary), (2-hydroxypropyl)-β-cyclodextrin (degree of substitution: 6.3; molecular weight: 1500.9 g mol^−1^) was purchased from Wacker Chemie AG (Munich, Germany). Simulated intestinal fluid (pH 6.8), without enzymes, was prepared based on the chapter 5.17.1 of the European Pharmacopoeia (10th edition). A 77.0 mL volume of 0.2 M NaOH, 250.0 mL of a solution containing 6.8 g of KH_2_PO_4_, and 500 mL of purified water were mixed, pH was adjusted to pH 6.8 and diluted to 1000 mL with purified water. Simulated gastric fluid (pH 1.2) was prepared as the following: 1 g of NaCl was dissolved in purified water and 80 mL 1 M HCl was added, finally, the solution was diluted to 500 mL with purified water. All materials used for this solution were purchased from Sigma-Aldrich (Budapest, Hungary).

### 2.2. Methods

#### 2.2.1. Preparation of Co-Ground Mixtures

Co-ground mixtures of TER and CD derivatives were prepared by grinding substances in a 1:1 molar ratio for 105 min in the case of both DIMEB and HPBCD. In this work agate mortar and pestle was chosen as a tool of preparation because it was adequate to produce the small quantities to be made.

To determine the exact weight ratio of the products, it is necessary to know the water content of the CDs. The moisture contents of the CDs were measured at 105 °C using a Mettler-Toledo HR73 (Mettler-Toledo Ltd., Budapest, Hungary) halogen moisture analyzer. The moisture content of DIMEB and HPBCD was 2.05 m/m% and 5.17 m/m%, respectively.

In both cases, 1 g of total product was prepared, which means that 0.1944 g of TER and 0.8056 g of DIMEB were measured. With the same calculation, the HPBCD product contained 0.1713 g of TER and 0.8287 g of HPBCD. Suitable quantities of samples (approximately 50 mg) were removed immediately after homogenization as a physical mixture (0 min) and at prescribed intervals (15, 30, 45, 60, 75, and 105 min) for further physicochemical evaluation.

The products were stored in vials with screwcap at 15–25 °C and relative humidity of 40–60% for the duration of the experiments.

#### 2.2.2. Phase Solubility Studies

The solubilizing potential and complexing tendencies of the two CD-derivatives (HPBCD and DIMEB) with TER in aqueous solution were evaluated using phase solubility methods described by Higuchi and Connors [26]. An excess amount of TER was added to aqueous solutions containing different concentrations of CDs, ranging between 0 and 50 mM. Solutions were stirred at 25 °C for 24 h in sealed flasks. Samples were filtered (0.22 μm pore size syringe membrane filter), then analyzed by Unicam UV/VIS spectrometer (Thermo Fisher Scientific, Waltham, MA, USA) at 284 nm to quantify the solubilized TER. The phase solubility diagram was constructed by plotting the dissolved TER against the concentration of CDs. The apparent stability constants (K_C_) of the complexes were calculated from the phase solubility diagram using the Higuchi–Connors equation:(1)$$K_{C}=\frac{slope}{S_{0}\left(1-slope\right)}$$
where *S*_0_ is the intrinsic solubility of TER.

#### 2.2.3. Thermal Analysis

The DSC analysis was performed with a Mettler-Toledo DSC 821e instrument (Greifensee, Switzerland), the heating rate was 5 °C min^−1^, in the temperature interval of 25–300 °C, argon was used as a carrier gas (10 L h^−1^). All samples and raw materials were characterized with these parameters, the investigated quantity was in the range of 2–5 mg, and examinations were performed in sealed Al pans of 40 μL with three leaks.

In addition, thermogravimetric analysis (TG) was performed with Mettler–Toledo TGA/DSC1 (Mettler–Toledo GmbH, Greifensee, Switzerland) instrument. Only HPBCD-containing samples were evaluated with this method. The weight of the samples ranged from 5.0 to 5.5 mg. Samples were heated from 25 to 300 °C with a heating rate of 10 °C min^−1^.

The evaluation of all measurements was performed with STAR VER 9.30 software for both DSC and TG experiments.

#### 2.2.4. X-ray Powder Diffractometry (XRPD)

For XRPD measurements, a BRUKER D8 Advance diffractometer (Karlsruhe, Germany) system was used with Cu Kα radiation (λ = 1.5406 Å) installed with a common sample changer and a VÅNTEC-1 line detector. The measurements were performed with Cu-KαI radiation at a wavelength of 1.5406 Å, X-ray tube voltage was 40 kV and current was 40 mA, in the interval of 3–30° 2θ. The obtained data were evaluated (Kα2-stripping, background removal, and smoothing) with DIFFRACplus EVA software (version 5.2).

The crystallographic changes caused by the temperature change were monitored with the same diffractometer equipped with an MRI Basic hot-humidity chamber (MRI Physikalische Geräte GmbH, Karlsruhe, Germany) controlled by an Ansyco Sycos H-Hot (Analytische Systeme und Componenten GmbH, Karlsruhe, Germany). HOT-XRPD measurements were carried out in the temperature range of 30–240 °C, in 5 °C increments. No change in humidity was applied during the experiments, the measurements were performed at the general room humidity.

#### 2.2.5. Fourier-Transform Infrared Spectroscopy (FT-IR)

Spectra were recorded on a Bio-Rad Digilab Division FTS-65A/896 FT-IR spectrometer, using Harrick’s Meridian SplitPea single reflection, diamond ATR accessory. Measurements were performed between 4000 and 400 cm^−1^, at an optical resolution of 4 cm^−1^, 256 scans were averaged to increase the signal-to-noise ratio.

Spectral manipulations were performed by using Thermo Scientific GRAMS/AI Suite software (version 9.0). Subtracting atmospheric water vapor superimposed on the sample spectrum was performed using a measured water vapor spectrum. Curve-fitting algorithm was applied with the Gaussian–Lorentzian function. The best curve-fitting procedure was achieved by iterative fits toward a minimum standard error. The change in the areas of the peaks was given as a percentage, it was calculated by comparing the area of the changing peaks to the sum of the areas of these peaks.

#### 2.2.6. Raman Spectroscopy

Raman spectra were recorded with a Thermo Fisher DXR Dispersive Raman spectrometer (Waltham, MA, USA) equipped with a CCD camera and a diode laser (λ = 780 nm). The following parameters were used during the measurements: the applied laser power was 12 and 24 mW at 25 µm slit aperture size; spectra were collected with an exposure time of 6 sec. The data were collected in the spectral range of 3300–200 cm^−1^ using automated fluorescence corrections. OMNIC for Dispersive Raman 8 software package (Thermo Fisher Scientific, Waltham, MA, USA) was used for data collection, averaging a total of 20 scans. For the removal of cosmic rays, a convolution filter was applied to the original spectrum using Gaussian kernel.

In addition to recording individual spectra, the Raman mapping method was also applied to obtain images on the distribution of the ingredient in the samples. Spectra were collected using a five times six grid from the 750 times 250 µm size area of the samples, defined by the optical microscope.

#### 2.2.7. Scanning Electron Microscopy

The morphological appearance and particle size characterization of the inclusion complexes were studied by scanning electron microscopy (Hitachi S4700, Hitachi Scientific Ltd., Tokyo, Japan) at an excitation voltage of 10 kV. The samples were previously coated with a thin (approximately 10 nm) gold-palladium film from a coater sputter (Bio-Rad SC 502, VG Microtech, Uckfield, UK). The particle size was obtained using the ImageJ 1.44p software (Bethesda, MD, USA), average diameter was calculated by selecting 50 particles randomly.

Statistical analysis was performed to determine the existence of a significant difference between the calculated data. For all characterized products, one-way analysis of variance (ANOVA) was performed with the post hoc Tukey HSD test. Likewise, the particle diameters of the samples produced by different preparation methods were examined. Statistically significant experimental results were assumed with *p* values < 0.05 and <0.01.

#### 2.2.8. In Vitro Dissolution Rate Studies

TER, physical mixtures and ground products were characterized by in vitro dissolution rate studies. Studies were performed using the rotating paddle method in decreased volume. The simulated intestinal and gastric medium prescribed in the pharmacopoeia is 900 mL and the marketed formulations containing 250 mg of TER, the dissolution medium, and the amount of active ingredient were reduced proportionally in our experiments. A total of 27.77 mg of TER, or a product containing 27.77 mg of TER, was measured, and added to 100 mL of each medium, the paddle was rotated at 100 rpm. Aliquots were withdrawn and replaced with fresh dissolution medium at given times (5, 10, 20, 30, 60, 90, and 120 min) and immediately filtered (syringe membrane filter with a pore size of 0.22 μm). After adequate dilution with the dissolution medium, the concentration of the dissolved drug was determined using a Unicam UV/VIS spectrometer (Thermo Fisher Scientific, Waltham, MA, USA) at 284 nm.

Cumulative dilution was calculated taking into account the volumes exchanged at sampling times. The obtained dissolution profiles were evaluated based on dissolution efficiency (DE) at 30, 60 and 120 min and mean dissolution time (MDT). For the determination of DE, the area under the dissolution curve up to a given time is related to the value of the area under the curve corresponding to the maximum (100%) dissolution up to the same time, which can be calculated by the following equation:(2)$$DE=\frac{\int_{0}^{t}ydt}{y_{100}}100\%$$
where *y* and *y*_100_ are the cumulative percentage dissolution at time *t* and at 100% dissolution [27].

The calculation of MDT considers the percentage of dissolved drug measured at two consecutive time points *t* and *t*_(*i*−1)_, thereby the time at which 63% of the API was dissolved can be determined.
(3)$$MDT=\frac{\sum_{i=1}^{n}t_{mid}\Delta M}{\sum_{i=1}^{n}\Delta M}$$
where *i* is the sample number, *n* is the number of dissolution time, *t_mid_* is the midpoint between the two *t* and *t*_(*i*−1)_, and Δ*M* is the additional amount of TER dissolved between *t* and *t*_(*i*−1)_ [28].

#### 2.2.9. In Vitro Diffusion Studies

Diffusion studies were performed to model the passive diffusion of TER across biological membranes. Tests were carried out in a modified horizontal diffusion model. The donor phase represented digestive fluids, and measurements were performed in both gastric and intestinal fluids. These were the same as those used in the dissolution rate studies. Phosphate buffer (pH 7.4) was used as the acceptor phase (representing blood) in all cases. Each phase contained 50 mL of liquid, kept at 37 °C and the rotation rate of the stir-bars was set to 300 rpm. A membrane (Whatman^TM^ regenerated cellulose membrane filter with 0.45 µm pores) with a diffusion surface of 2.14 cm^2^ was soaked in isopropyl myristate for 30 min and placed between the two phases. The pure TER, physical mixtures and 120 min ground products were characterized by the method. Every measured sample contained 13.885 mg of TER and was placed into the donor phase. All measurements were performed for 120 min.

The diffused drug concentration of the acceptor phase was measured spectrophotometrically by an AvaLight DH-S-BAL spectrophotometer (AVANTES, Apeldoorn, The Netherlands), which was connected to an AvaSpec-2048L transmission immersion probe (AVANTES, Apeldoorn, The Netherlands) allowing real time measurements at the wavelength of 284 nm. The apparent permeability coefficient (*P_app_* (cm s^−1^)) was calculated for the samples and TER according to Equation (4):(4)$$P_{app}=\frac{Q}{A\;c\;t}$$
where *Q* is the cumulative amount of drug permeated (µg), *A* is the surface of the diffusion area of the artificial membrane (cm^2^), *c* is the initial concentration of the drug in the donor phase (µg cm^−3^), and *t* is the total time of the experiment (s). The permeation enhancement ratio (*R*) was calculated to compare products to the pure drug according to Equation (5):(5)$$R=\frac{P_{\mathrm{app}}\left(\mathrm{sample}\right)}{P_{\mathrm{app}}\;\left(\mathrm{control}\right)}$$
where *P_app_* (sample) and *P_app_* (control) are the apparent permeability coefficient (cm s^−1^) of products and TER, respectively. The results measured every five minutes were considered to calculate these values.

## 3. Results and Discussion

### 3.1. Phase Solubility Studies

Figure 1 shows the increased solubility of TER in the presence of increasing CD concentration, the diagrams displaying straight lines with a slope less than 1. The diagrams are Higuchi A_L_ type for both CDs, therefore the formation of 1:1 API:CD molecular ratio can be assumed. The apparent solubility constant (K_C_) was calculated from Figure 1 using the Higuchi–Connors equation, and it was found to be 139.93 M^−1^ in the case of DIMEB and 96.93 M^−1^ in the case of HPBCD. The higher stability constant indicates that TER has a higher affinity for DIMEB than for HPBCD.

### 3.2. Thermal Analysis

Figure 2A shows the DSC thermograms of TER, DIMEB, and all samples taken during the co-grinding process with these materials, while Figure 2B presents the same scheme in the case of products containing HPBCD. At low temperatures, a broad endothermic signal (between 25–85 °C) is observed for both CDs, indicating the presence of water in the CD derivative. DIMEB also shows an exothermic peak at 187 °C, indicating crystallographic phase transition. DSC data showed a sharp peak of the TER melting point near 209 °C, and a complex phenomenon at a higher temperature range, suggesting the degradation of TER. These peaks do not overlap, thus thermoanalytical changes related to the process can be detected. DIMEB-containing products showed a complex thermoanalytical signal at a temperature range below the melting point of the API, this peak broadens with increasing grinding time, in the final product it can be detected at 150–190 °C. In the case of HPBCD, this phenomenon was not observed, but between 220 and 260 °C, a broad complex endothermic peak appeared, which is probably related to the degradation process of the active substance. TG measurements were performed to explain the complex thermoanalytical signal of the HPBCD products in the range of 220–260 °C. The mass of the sample showed a rapid decrease in this temperature range, suggesting the decomposition of the API in these products (data not shown).

No characteristic melting point of TER was detected in any of the co-ground products, which may indicate rapid in situ complexation during the measurement. However, DSC measurements indirectly prove the complexation between the components because the decrease of the melting point of TER can be associated with the amorphization of API. XRPD and FTIR measurements can be used to clarify the phenomenon.

### 3.3. X-ray Powder Diffractometry

Among the materials used, only the diffractogram of TER contained sharp peaks indicating crystallinity, CDs being amorphous materials do not exhibit well-defined peaks in their diffractograms. The same changes can be seen for both CDs. Physical mixtures showed the characteristic peaks of TER, the intensity of which was proportionally lower due to the weight ratio of the mixture used. The intensity of these peaks decreased as the grinding time progressed, and after 75 min of the process, the diffractograms showed no well-defined peaks. Thus, it can be stated that products contained TER in an amorphous phase in the case of both CDs. To investigate the effect of grinding after this point, the process was conducted until 105 min. However, no significant change was observed in the diffractograms between the 75- and 105-min products (Figure 3).

### 3.4. Hot-Stage X-ray Powder Diffractometry (HOT-XRPD)

To complement the phenomena seen in the DSC thermograms, XRPD measurements were performed at increasing temperatures, in a hot-stage chamber. Measurements were performed with 105-min co-ground products.

At room temperature, the ground products appeared amorphous. However, due to heat transfer a multi-step crystallization process takes place in DIMEB containing products starting at 135 °C. The most intense peaks in the first part are at 175 °C. The peaks in the second stage are different from the peaks detected at 175 °C, which shows the appearance of a second new crystalline phase. Further heating melted the product completely at about 240 °C. Figure 4 shows the selected curves, the same process can be seen under the curves in 2D visualization. The characteristic peaks that appeared were not identical to those of TER. Furthermore, there is no evidence in the literature that TER exhibits polymorphism. This information allows us to conclude that in this wide temperature range a new crystalline phase forms. This change and temperature range are comparable with the exothermic peak between 135 °C and 190 °C. The differences in temperature ranges are due to the fact that the measurement lasted longer at a given temperature during HOT, thus more heat was transferred to the sample.

To further investigate the phenomenon, both TER and DIMEB were ground for 105 min and HOT-XRPD measurements were performed under the same conditions. In neither case were there any new peaks indicating recrystallization (data not shown).

In the case of HPBCD, no such phenomena were detected. Heating the product with the same parameters did not change the amorphous property. Furthermore, cooling the product did not result in a crystalline material, referring to a stable amorphous property (data not shown).

### 3.5. Vibrational Spectroscopy

Vibrational spectroscopic methods—infrared and Raman spectroscopies—are especially useful tools in the investigation of amorphous materials. Although the information provided by the methods is based on the same set of normal vibrations, they supplement each other due to the difference in the general selection rules determining the intensities of the peaks in the spectra. The intensities of the peaks in the IR spectra originate from the alteration of the electric dipole moment during the normal vibrations, while the Raman intensities depend on the magnitude of the alteration of the polarizability tensor caused by the normal vibrations [29]. The combination of these methods can be very useful in the case of materials such as TER, which contain either charged or highly polarized parts and non-polar, but electron-rich parts such as multiple bonds and aromatic rings.

The high energy part of the infrared spectrum (Figure 5A(I)) shows the N-H stretching band of the tertiary ammonium ion around 2440 cm^−1^ [30], while that of the Raman spectrum (II) shows the stretching mode of the triple carbon–carbon bond at 2327 cm^−1^ [31]. Similarly, a practically total exclusion rule can be identified in the other parts of the spectra (Figure 5B,C) except in cases such as the peak of the stretching mode of the carbon–carbon double bond at 1636 cm^−1^ [29], which proves that there is no symmetry centrum in the crystal structure. The relative intensity of these peaks helps to identify the part of the molecule and the symmetry of the normal mode they belong to.

The infrared spectra of the samples obtained after different grinding times showed well-recognizable alterations. The broad ammonium N-H stretching band smoothed into the baseline after 30 min, and most of the characteristic bands of TER also disappeared, broadened, and melted into the broad bands of the product, either in the case of DIMEB or HPBCD. The only region where the advance of the formation of the product was possible to follow for a longer period of time was the region of the out-of-plane vibrations of the aromatic rings (Figure 6) [32].

The traces of even these bands characteristic on crystalline TER disappeared between 75 and 105 min of grinding in both cases.

This range of the spectra allowed us to quantify the advance of the process by applying the peak fitting method to the bands in this region. The results are shown in Figure 7. The iterations took place with the fewest possible fitted parameters until the standard error dropped below 4∙10^−4^, and the residual curve was checked for noise-like behavior. The kinetic curves were constructed from the contributions of the areas of those peaks which increased and the areas of those peaks which decreased their intensity in the sum of their areas. The kinetic curves showed the shape of the concentration functions of the autocatalytic processes [33], which is known in the phase transition field as Avrami’s model [34,35].

There is a disadvantage of Raman microscopy when it is applied for the characterization of a heterogeneous sample. None of the recorded spectra represented the overall composition of the sample [36], thus another approach was used to study the changes of our samples in time. The advantage of the method is that it can record the spectra on a macroscopic part of the heterogeneous sample, selected through the optical microscope according to a predefined grid [37].

Intensity maps of various peaks can be constructed to monitor the distribution of the components in the sample. The success of this method depends on the proper selection of peaks for the construction of maps. It is necessary to find peaks that are characteristic only of the product and which are only on the starting materials. Unfortunately, there was no proper peak found for the CDs employed, due to their broad and weak bands, which strongly overlapped with the similar features of the products. On the other hand, the narrow peaks of crystalline TER assured more probable success. The region of the stretching bands of the triple and the double bond might have provided good possibilities, but the products also displayed some intensities at those wavenumbers (Figure 8A,B). Unfortunately, the range of the out-of-plane deformation modes of the aromatic rings was also blocked by the broad bands of the products (Figure 8C). There was only one peak, unexpectedly in the difficult-to-assign fingerprint region, at 1289.5 cm^−1^, which proved to be useful to monitor the presence and the distribution of crystalline TER in the sample (Figure 8D).

All samples were mapped using the peak at 1289.5 cm^−1^ and they also confirmed that crystalline TER disappeared from the samples after 75 min of grinding, and it was found in “islands” in the samples of earlier stages of grinding (see sample maps in Figure 9).

Alterations in the vibrational spectra reflect the changes of the state of the components in the samples, thus the comparison of all vibrational spectra recorded should allow us to arrive at a conclusion regarding the most probable method of formation of the inclusion complex which formed during the grinding process. In order to judge the changes in the IR spectrum, the following procedure was performed. All spectra, which were recorded at the various stages of the grinding process, were averaged. Since these spectra, one for TEB—DIMEB, the other for the TER—HPBCD mixtures, contained the peaks of the starting materials above those of the product, careful spectral subtraction was conducted to remove them. The result is shown for the TER—DIMEB mixture in Figure 10.

Spectra resulted from the successive spectral subtractions, mainly resembling the spectra of CDs. None of the bands could be assigned to the TER-part of the product, except in the range of the out-of-plane deformation bands of the α-naphthalene group (Figure 10D). Three bands from 808, 792, and 776 cm^−1^ shifted to 809, 798, and 778 cm^−1^, respectively. The extent of these shifts was insufficient to deduce that the interaction between the components took place through the aromatic part of TER. On the other hand, the missing bands of the unsaturated chain and the tertiary ammonium group allowed the conclusion that inclusion complexes were formed, in which the whole chain was hidden in the non-polar internal part of CDs.

### 3.6. Scanning Electron Microscopy

SEM measurements were performed to describe morphology and particle size, these results are shown in Figure 11 and Figure 12. The results of SEM analysis were consistent with those observed with XRPD and DSC. TER crystals appear as clumped in clusters, with a smooth surface without cracks, showing a flat crystalline profile, with an average diameter of 13.44 µm. CD-containing products showed different morphology, the crystal of TER was no longer detectable in either of the two products, where the formation of amorphous aggregates was observed.

The particle size of the compositions decreased with grinding for both DIMEB and HPBCD, where the mean diameters were 1.44 µm and 3.35 µm, respectively. These were significantly different from the active substance (*p* < 0.01). However, the difference between the two products was also significant (*p* < 0.05) in this respect.

### 3.7. Dissolution Studies

Dissolution studies were carried out by paddle method in decreased volume of dissolution medium. The dissolution profiles were plotted as a percentage of dissolved drug as a function of time. 

The solubility of the active ingredient is strongly pH-dependent, as it is significantly lower at higher pH. In the simulated intestinal fluid, the solubility of TER is below 1% and the solubility-enhancing effect of CDs is evident with both DIMEB and HPBCD. However, this increase was diverse depending on the excipient used (Figure 13). The DIMEB-containing product increased DE to 4.81% by 120 min, according to the MDT value, the dissolution rate was higher. However, the DE of the HPBCD product rose to a lesser extent to 1.4%, and this was reached with a lower dissolution rate compared to the pure drug (Table 1). During the measurements, the solubility of TER reached almost 100% in simulated gastric medium, but CDs had a positive effect in this case as well, as they increased the dissolution rate of the API. This is shown by the decreased MDT values (Table 1). Data regarding physical mixtures are in Appendix A.

This improvement in solubility properties can be attributed to the fact that during the co-grinding process the average diameter of particles decreased, meanwhile, real complexes were prepared, and due to the fact that the solubility of the amorphous materials is many times higher than that of the crystalline materials. The difference in solubility increases between the two CDs can be attributed to the different particle size and different K_C_ values. Since TER has a higher affinity for DIMEB, we were able to achieve a better solubility enhancing effect.

### 3.8. In Vitro Diffusion Studies

The results of the diffusion studies are shown in Figure 14, which shows the amount of drug diffused as a function of time. The *P_app_* and enhancement ratio values calculated from the data obtained are given in Table 2. Data regarding physical mixtures are in Appendix A.

In the simulated intestinal fluid, the *P_app_* of TER was two orders of magnitude smaller than in the lower pH medium. The formulation containing DIMEB showed similar diffusion in the simulated intestinal medium, with the enhancement ratio near 1, and higher diffused drug in the simulated gastric medium, where the enhancement ratio was 2.232.

Surprisingly, diffusion was lower in the formulation containing HPBCD compared to the drug. This permeability decreasing effect of HPBCD was observed and studied previously [38]. The permeability-reducing effect may be as much a consequence of the host–guest interaction as of the solubility-enhancing effect. By complexing the lipophilic drug, cyclodextrin also reduces the free drug fraction. In turn, the reduced free fraction results in a reduced concentration gradient, thereby reducing the thermodynamic driving force for membrane permeation [39].

## 4. Conclusions

In the present work, we employed comprehensive analytical methods to evaluate the co-grinding method and to develop a fast-screening protocol required for the process. For this, a model drug and two different CD derivatives were chosen. Physicochemical characterization was used to follow through complexation. Using XRPD and DSC measurements, we were able to determine that the alleged complexes exhibit amorphous properties; however, the complexation was demonstrated by vibrational spectroscopy (FTIR and Raman spectroscopy) methods. The in vitro characterization of the complexes and their comparison with the API was performed by dissolution and diffusion studies.

## Figures and Tables

**Figure 1 pharmaceutics-14-00744-f001:**
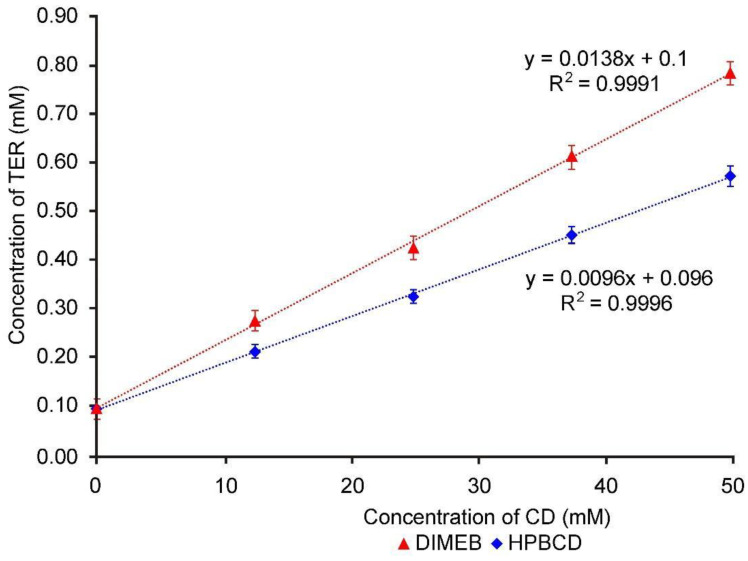
Phase solubility diagrams of TER in aqueous solutions containing DIMEB and HPBCD at 25 °C.

**Figure 2 pharmaceutics-14-00744-f002:**
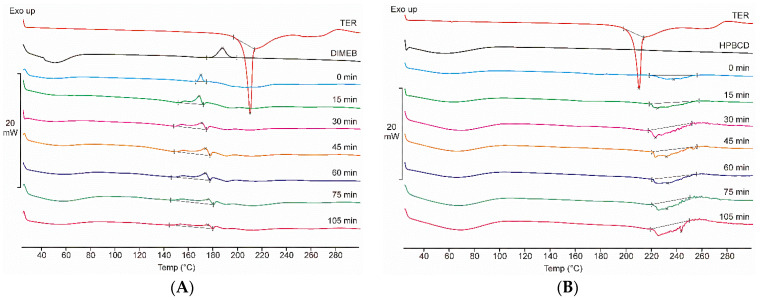
DSC thermograms of raw materials, physical mixture (0 min), and products (15–105 min) containing DIMEB (**A**) and HPBCD (**B**). In the case of DIMEB products, an exothermic peak appears before the melting point of the active substance. In the case of HPBCD products, a complex endothermic peak appears at higher temperatures.

**Figure 3 pharmaceutics-14-00744-f003:**
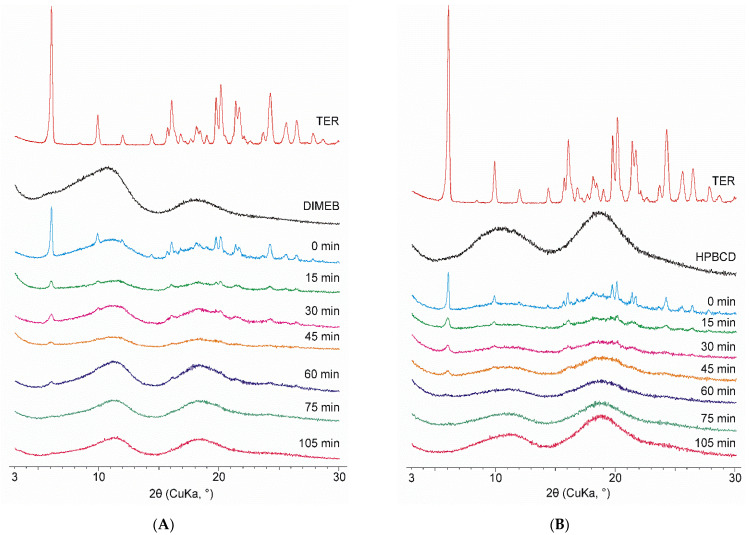
XRPD diffractograms of DIMEB (**A**) and HPBCD (**B**) samples taken during co-grinding (15–105 min) with starting materials and physical mixture (0 min).

**Figure 4 pharmaceutics-14-00744-f004:**
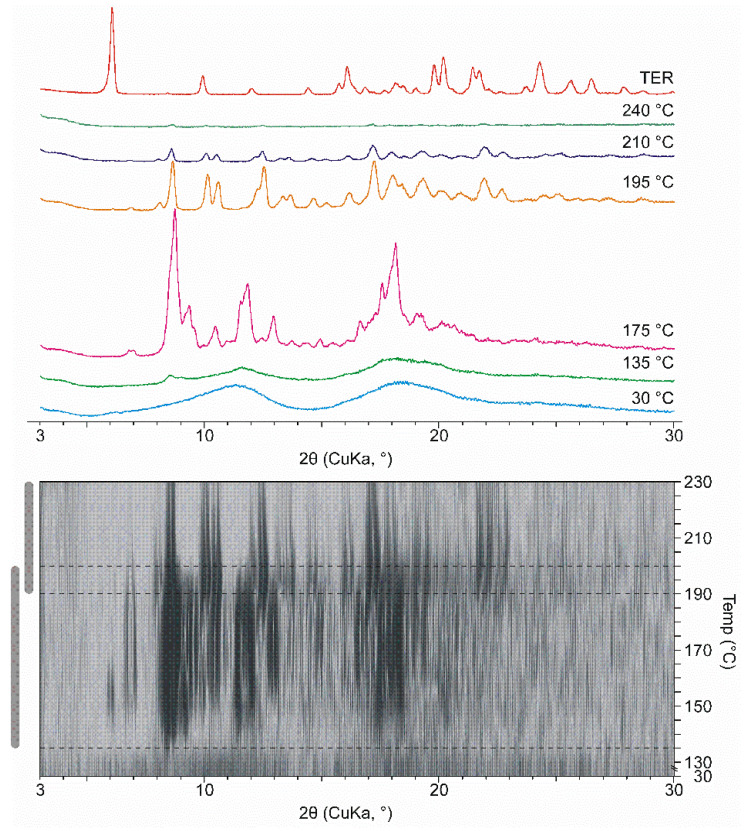
XRPD diffractograms were measured in various temperatures in the case of DIMEB product with 2D visualization. 2D visualization shows the temperature range and overlap of the two recrystallization phases.

**Figure 5 pharmaceutics-14-00744-f005:**
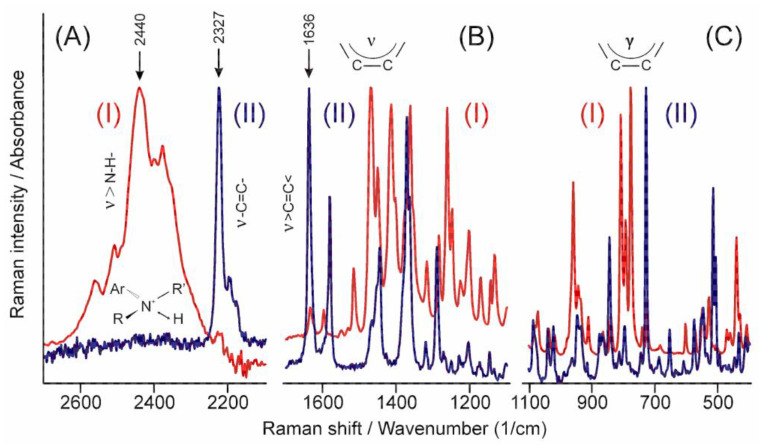
Comparison of the infrared (**I**) and the Raman (**II**) spectra of TER in certain characteristic regions. (**A**) stretching bands of +N-H and -C≡C- groups, (**B**) stretching bands of >C=C< group and the aromatic rings, (**C**) the out-of-plane deformation bands of the aromatic rings.

**Figure 6 pharmaceutics-14-00744-f006:**
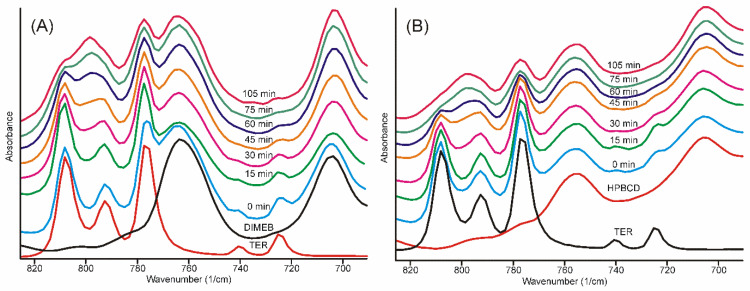
FTIR spectra of samples. The evolution of the bands in the out-of-plane deformation vibrations of the aromatic rings as the grinding time increases. The characteristic bands of TER (**A**) DIMEB and (**B**) HPBCD practically disappeared between 75 and 105 min.

**Figure 7 pharmaceutics-14-00744-f007:**
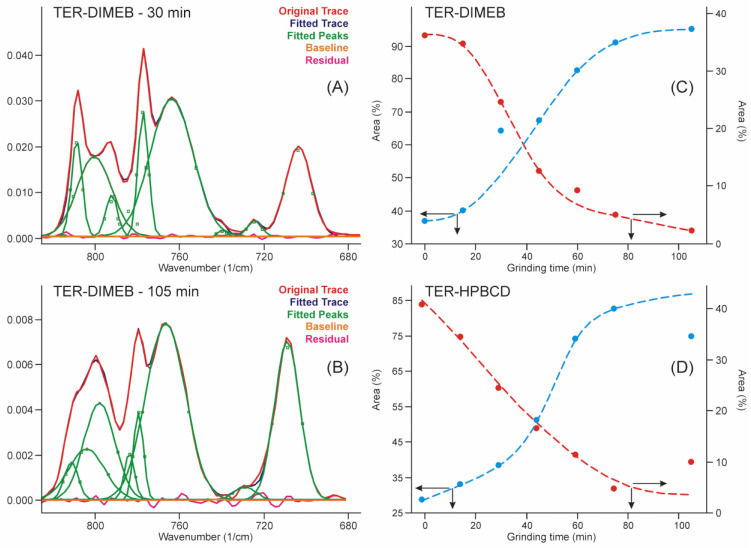
Peak fitting of FTIR spectra in the range of the out-of-plane deformation region resulted in a sigmoid type of curves with the grinding time, which confirmed the expected autocatalytic phase transformation type of kinetics of the product formation. (**A**)—fitted curves after 30 min grinding with residues of the starting materials in the TER-DIMEB mixture, (**B**)—fitted curves after 105 min grinding without residues of the starting materials in the TER-DIMEB mixture. Kinetic curves for the starting materials and the products constructed from the areas of the fitted peaks (**C**)—in the case of TER-DIMEB, and (**D**)—in the case of TER-HPBCD mixture. The kinetic curves were calculated from the contribution of the areas of the peaks of increasing and those of decreasing intensities to the sum of the peak areas.

**Figure 8 pharmaceutics-14-00744-f008:**
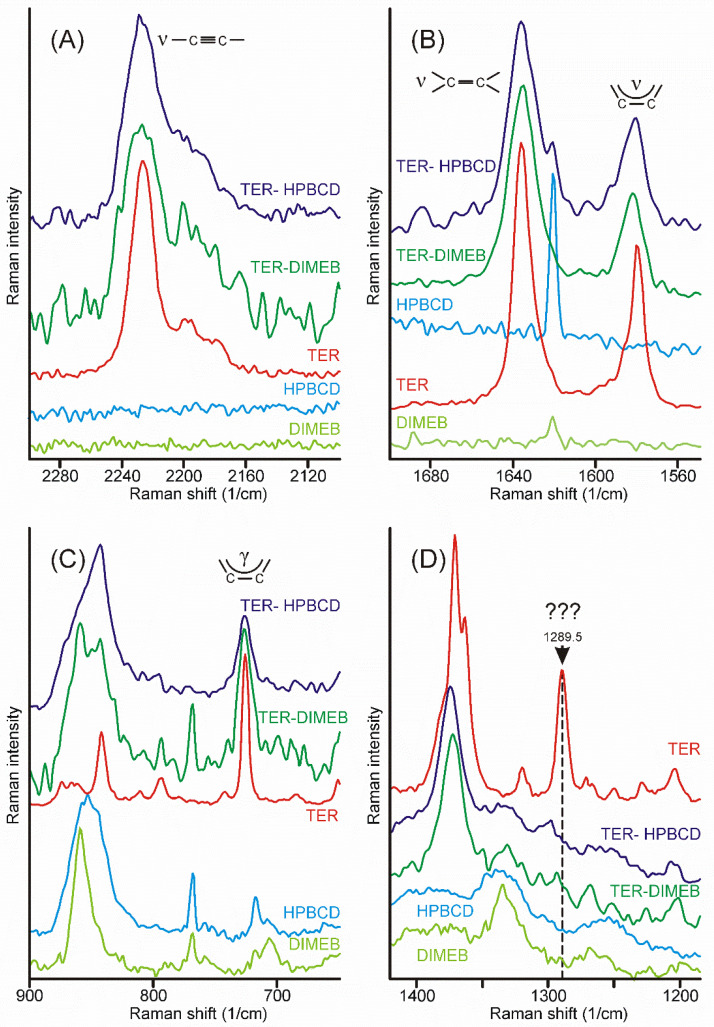
Most of the characteristic peaks in the Raman spectra of crystalline TER overlap with the bands of the products, except the one at 1289.5 cm^−1^, in the fingerprint region. (**A**)—region of the stretching band of the carbon–carbon triple bond, (**B**)—region of the stretching bands of the carbon–carbon double bond and those of the aromatic rings, (**C**)—region of the out-of-plane deformation bands of the aromatic rings, (**D**)—region containing the useful peak for the characterization of the presence and the distribution of crystalline TER in the samples.

**Figure 9 pharmaceutics-14-00744-f009:**
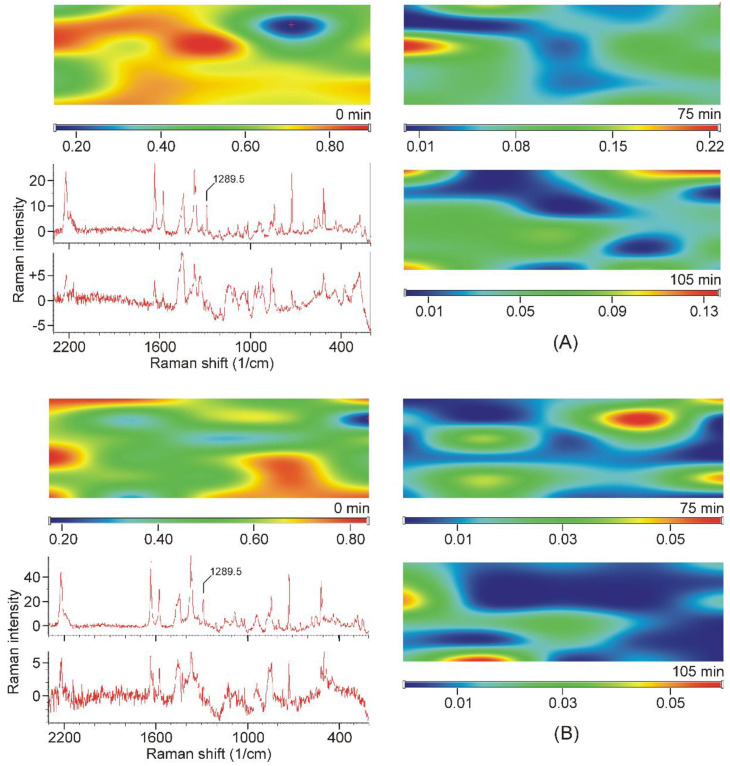
Examples of Raman-intensity maps constructed from the Raman spectra recorded at the grid points, using the intensities at 1289.5 cm^−^^1^ of the TER-DIMEB (**A**) and the TER-HPBCD mixture (**B**). Top left corner: Mixture of the starting materials. On the right: The maps of the samples taken in the final stage of grinding. Bottom left: Upper spectrum recorded in the high-intensity area of the map of the non-grinded mixture. Lower spectrum recorded at the low-intensity area of the map of the sample ground for 75 min (position marked with a red cross).

**Figure 10 pharmaceutics-14-00744-f010:**
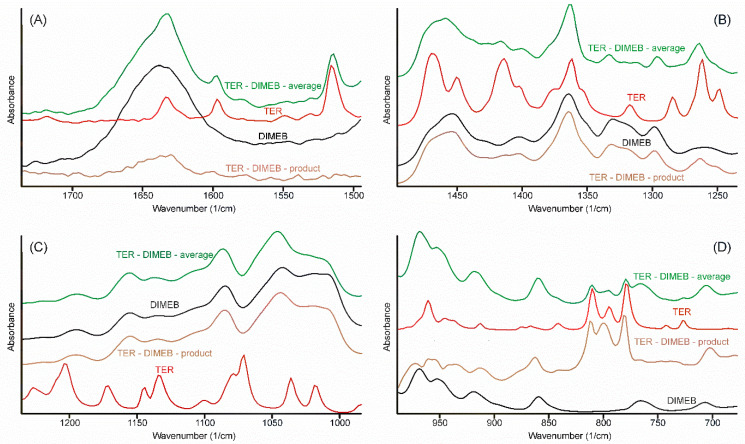
Construction of the FTIR spectrum (regions 1750–1495; 1495–1230; 1430–990; és 990–675 cm^−1^ (**A**–**D**) respectively) of the product in the TER-DIMEB system, by successive subtraction of the spectra of the starting materials from the average spectrum of the samples taken at various times of the grinding process. The spectrum of the product resembled the spectrum of DIMEB. Only the out-of-plane deformation bands of TER can be seen at 809, 798, and 778 cm^−1^, somewhat shifted. All other bands of TER disappeared, including the tertiary ammonium band (not shown in the Figure).

**Figure 11 pharmaceutics-14-00744-f011:**
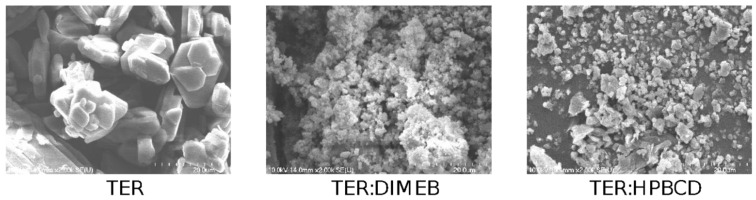
SEM images of TER and co-ground products at 2.00 k magnitude.

**Figure 12 pharmaceutics-14-00744-f012:**
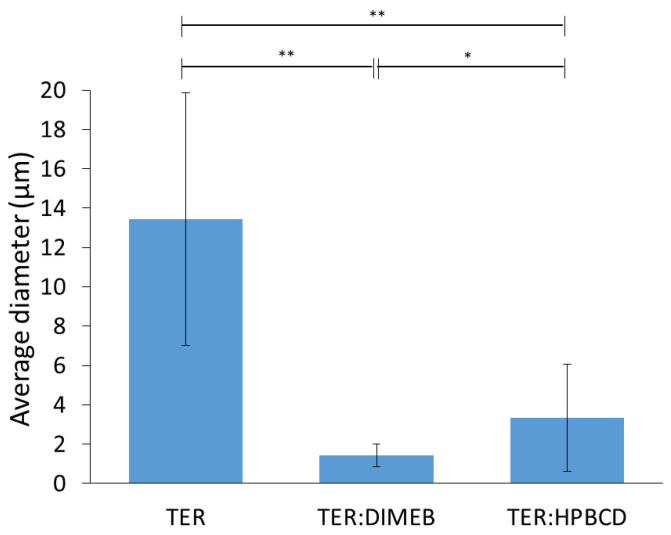
Average diameter and standard deviation of particles (** *p* < 0.01; * *p* < 0.05).

**Figure 13 pharmaceutics-14-00744-f013:**
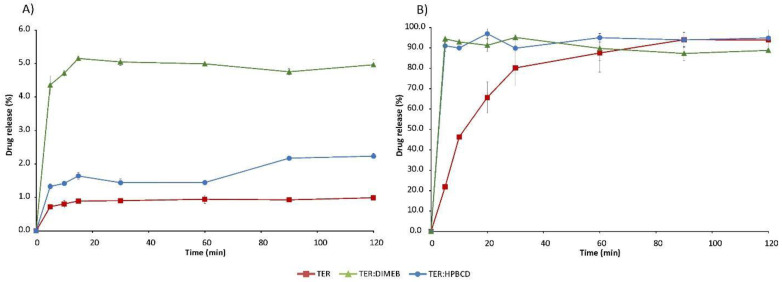
In vitro dissolution studies of TER and products in simulated intestinal fluid (**A**) and simulated gastric fluid (**B**).

**Figure 14 pharmaceutics-14-00744-f014:**
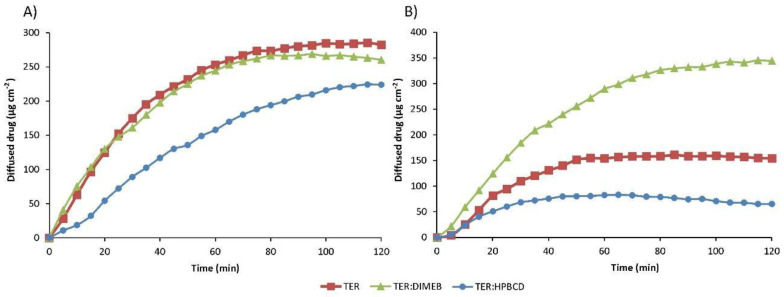
In vitro diffusion studies of TER and products in simulated intestinal fluid (**A**) and simulated gastric fluid (**B**).

**Table 1 pharmaceutics-14-00744-t001:** DE and MDT values with standard deviation in simulated intestinal and gastric media.

	Simulated Intestinal Medium				Simulated Gastric Medium			
	DE_30min_	DE_60min_	DE_120min_	MDT	DE_30min_	DE_60min_	DE_120min_	MDT
TER	0.78 ± 0.03	0.85 ± 0.05	0.90 ± 0.04	18.88 ± 7.49	53.30 ± 5.63	68.57 ± 6.91	80.45 ± 5.19	18.35 ± 2.69
TER:DIMEB	4.49 ± 0.10	4.75 ± 0.09	4.81 ± 0.09	11.02 ± 2.00	84.94 ± 0.82	88.68 ± 1.50	91.54 ± 1.02	12.54 ± 6.08
TER:HPBCD	1.37 ± 0.06	1.40 ± 0.04	1.70 ± 0.02	28.86 ± 1.63	85.71 ± 1.53	88.80 ± 0.84	88.54 ± 1.32	7.69 ± 5.68

**Table 2 pharmaceutics-14-00744-t002:** Apparent permeability coefficient (*P_app_*) and enhancement ratios (R).

	Simulated Intestinal Medium		Simulated Gastric Medium	
	*P_app_* (cm s^−1^)	R	*P_app_* (cm s^−1^)	R
TER	1.41 × 10^−4^		7.72 × 10^−6^	
TER:DIMEB	1.3 × 10^−4^	0.921	1.72 × 10^−4^	2.232
TER:HPBCD	1.12 × 10^−4^	0.793	3.28 × 10^−5^	0.425

## Data Availability

The datasets used and/or analyzed are available from the corresponding author on reasonable request.

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
