# Peer review of "Development of Solvent-Free Co-Ground Method to Produce Terbinafine Hydrochloride Cyclodextrin Binary Systems; Structural and In Vitro Characterizations"

_pharmaceutics, 2022, doi:10.3390/pharmaceutics14040744_

Round 1

Reviewer 1 Report

The manuscript is extremely well written and the studies well designed. I did not find any shortcomings on the design or data. The work is not very innovative per se, but it's executed very well and thoroughly.

Author Response

Reply to Referee comments

Reviewer 1

Thank you for taking the time and effort to carefully check and review our manuscript. We also appreciate your positive overall opinion of our work.

Reviewer 2 Report

Dear Authors,

Work is very interesting and well witten.

You have ha cleary ontresting findings about complex terbinafin and cyclodextrons.

IYou shoul clear out why NaCl was used in pH 1.2 and not pH 6.8 since ionice strength might

Iinfluence solubility and dissolution rate. Osmolality should be ajusted for intestinal simulated fluid.

AAlso modified dissolution paddle should be defined. Since it is not clear what is modification, and this us important

In order to define dissolution results.

Best regards

Author Response

Reply to Referee comments

Reviewer 2

Thank you for taking the time and effort to carefully check and review our manuscript. We also appreciate your positive overall opinion of our work. Below are the answers for your questions whereas modifications on the manuscript are highlighted by green colour.

You shoul clear out why NaCl was used in pH 1.2 and not pH 6.8 since ionice strength might influence solubility and dissolution rate. Osmolality should be ajusted for intestinal simulated fluid.

            Thank you for your remarks, your insights about the topic are right. However, in this study, we used simulated fluids described in European Pharmacopeia. We did not intend to change this, as the pharmacopeial guidelines should be taken into account for trials for which we have established methods in the pharmacopoeia. We did not intend to investigate the pH-dependent dissolution, rather simulating the expected dissolution in gastric and intestinal environment.

AAlso modified dissolution paddle should be defined. Since it is not clear what is modification, and this us important in order to define dissolution results.

            Thank you for your comment. The modification in this case meant the reduction of the volume of the dissolution medium. It is written in the method part, but based on your valuable comment, we modified the text slightly in this part and the results, to be easier to interpret.

Reviewer 3 Report

The submitted manuscript by B. A. Kondoros and coworkers describes a study to prepare complexes between beta-cyclodextrin derivatives (DIMEB and HPBCD) and Terbinafine hydrochloride (as a model drug) via co-grinding. The solid products of the co-ground mixtures were analyzed by diverse analytical techniques such as XRPD, DSC, FT-IR, Raman spectroscopy and SEM. Also, in vitro solution studies were performed like phase solubility, dissolution, and permeation.

I agree with authors that a green technology to prepare pharmaceutical formulations is highly desirable, and that solvent-free co-grinding process is a tool with potential to fulfill this goal. In fact, there is already a large body of literature published in this topic with cyclodextrins, which has been recently summarized by P.A. Mura and coworkers in Pharmaceutics 2018, 10, 189 (doi:10.3390/pharmaceutics10040189). In the cited review article is acknowledged that “the methods able to directly demonstrate an actual inclusion complex formation in the solid state are limited” and that “DSC and XRPD data are able only to infer the inclusion phenomena in the analyzed sample” (pp. 4-5, Ref. 18).

Thus, despite the large number of studies presented herein, the authors are taking the common approach of comparing thermal, spectral and morphological properties of the putative complex prepared by grinding and that of the pure compounds. However, two very important pieces of information are still missing: i) a comparison with the drug-CD physical mixture and ii) direct experimental proof that an inclusion complex is indeed formed in the solid-state. None of the studies performed are well-suited to demonstrate the formation of the inclusion complex in the solid-state. In this regard, I appreciated the efforts in trying to link the alterations in the vibrational spectra with the possible inclusion complex. However, the simple disappearance of some bands is not direct proof of formation of the inclusion complex. Also, the conversion of crystalline drug to amorphous during milling could be also a consequence of numerous other phenomena like for instance the formation of a solid dispersion.

In my view, the current manuscript is not yet acceptable for publication given the lack of novelty (both conceptually and non-conclusive data) and missing of a direct proof of the formation of the inclusion compound. I highly recommend comparing results with the drug-CD physical mixture and to incorporate solid-state NMR experiments to demonstrate the formation of the inclusion compound in the solid phase. Otherwise, conclusions (and even the title of the article) are not supported.

Author Response

Reply to Referee comments

Reviewer 3

Thank you for taking the time and effort to carefully check and review our manuscript. We also appreciate your positive overall opinion of our work. Below are the answers for your questions.

However, two very important pieces of information are still missing: i) a comparison with the drug-CD physical mixture

            Thank you for your remarks.

  1. The drug-CD physical mixture (PM) is listed in the manuscript as a 0-minute ground product. Most of the experiments performed include comparisons with physical mixtures. Probably we were not clear enough, but the first measurements, at zero minutes were taken with the physical mixtures of TER and the corresponding CD. It is stated in Chapter 2. Materials and Methods.

“Suitable quantities of samples (approximately 50 mg) were removed immediately after homogenization as a physical mixture (0 min) and at prescribed intervals (15, 30, 45, 60, 75, and 105 min) for further physicochemical evaluation.” – page 3.

Only the in vitro dissolution and diffusion studies lack these comparative measurements. Based on your suggestion, we have performed these measurements. Although we did not intend to increase the scope of the article with these data, since editorial decision was minor revision. Table 1 contains the dissolution efficiency (DE, %) at 120 min and MDT values. DE of PMs are higher than pure drug, due to the solubilizing effect of dissolved CDs, however lower than ground products.

Products

% DE 120 min

MDT

Simulated intestinal medium

TER

0.90

18.88

TER:DIMEB_PM

4.11

12.84

TER:HPBCD_PM

1.27

12.12

TER:DIMEB

4.81

11.02

TER:HPBCD

1.70

28.86

Simulated gastric

medium

TER

80.45

18.35

TER:DIMEB_PM

74.91

9.13

TER:HPBCD_PM

84.87

10.43

TER:DIMEB

91.54

12.54

TER:HPBCD

88.54

7.69

Table 1. Calculated dissolution efficiency (% DE) at 120 min and MDT values of pure drug, physical mixtures (PM) and co-ground products.

Table 2. shows the calculated apparent permeability coefficient (Papp) and enhancement ratios (R). In case of DIMEB-products, co-ground products and PMs showed similar permeability. However, diffusion from HPBCD containing PMs decreased from intestinal medium, the diffused drug was undetectable under the measurement conditions. On the other hand, the drug diffused from gastric juice resulted in a higher concentration in the acceptor phase than the ground products.

Products

Papp

R

Simulated intestinal medium

TER

1.41E-04

-

TER:DIMEB_PM

1.34E-04

0.950

TER:HPBCD_PM

-

-

TER:DIMEB

1.30E-04

0.921

TER:HPBCD

1.12E-04

0.793

Simulated gastric

medium

TER

7.72E-06

-

TER:DIMEB_PM

1.76E-06

2.280

TER:HPBCD_PM

1.86E-06

2.422

TER:DIMEB

1.72E-04

2.232

TER:HPBCD

3.28E-05

0.425

Table 2. Calculated apparent permeability coefficient (Papp) and enhancement ratios (R) of pure drug, physical mixtures (PM) and co-ground products.

I agree with authors that a green technology to prepare pharmaceutical formulations is highly desirable, and that solvent-free co-grinding process is a tool with potential to fulfill this goal. In fact, there is already a large body of literature published in this topic with cyclodextrins, which has been recently summarized by P.A. Mura and coworkers in Pharmaceutics 2018, 10, 189 (doi:10.3390/pharmaceutics10040189). In the cited review article is acknowledged that “the methods able to directly demonstrate an actual inclusion complex formation in the solid state are limited” and that “DSC and XRPD data are able only to infer the inclusion phenomena in the analyzed sample” (pp. 4-5, Ref. 18).

                Based on the second paragraph of Referees 3. opinion, in which one of Prof. Mura’s reviews was quoted. Unfortunately, in the quoted article Ref 18. cannot be found on pages 4-5. The correct reference is Ref. 16., which is another review article of Prof. Mura (P. Mura, Analytical techniques for characterization of cyclodextrin complexes in solid state: A review, J. Pharm. Biomed Anal., 113, (2015) 226-238.).

Unfortunately, we feel that it is necessary to disagree with the interpretation of the content of those articles with Reviewer 3. Ref. 16., clearly states in its abstract, that “The analytical characterization of drug–cyclodextrin solid systems and the assessment of the actual inclusion complex formation is not a simple task and involves the combined use of several analytical techniques, whose results have to be evaluated together.” It was just the approach, which we used during our work!

Furthermore, the sentence, originally quoted by Reviewer 3., has the same meaning. So, asking for “direct experimental proof that an inclusion complex is indeed formed in the solid state” seems to be asking for the impossible.

  1. ii) direct experimental proof that an inclusion complex is indeed formed in the solid-state. None of the studies performed are well-suited to demonstrate the formation of the inclusion complex in the solid-state. In this regard, I appreciated the efforts in trying to link the alterations in the vibrational spectra with the possible inclusion complex. However, the simple disappearance of some bands is not direct proof of formation of the inclusion complex. Also, the conversion of crystalline drug to amorphous during milling could be also a consequence of numerous other phenomena like for instance the formation of a solid dispersion.

I highly recommend comparing results with the drug-CD physical mixture and to incorporate solid-state NMR experiments to demonstrate the formation of the inclusion compound in the solid phase.

The last paragraph in the review mentions the solid-state NMR (SSNMR) as a method which is capable of fulfilling this task, but it is a bit overestimation of its capabilities, since this is also a spectroscopic method, where the measured data needs interpretation, like in the case of other spectroscopic methods, e.g. vibrational spectroscopies. Furthermore, the interpretation of the results of an SSNMR measurement requires a more complex approach than those of a solution one. See the following articles as examples: Brend Reif, S.A. Ashbrook, L. Emsley, and Mei Hong, Solid-state NMR spectroscopy, Nature Reviews, Methods Primers, Art. ID (2021) 1:2.  and
R.T. Berendt, D.M. Speger, P.K.Isbester, E.J. Munson, Solid-state NMR spectroscopy in pharmaceutical research and analysis, Trends in Analytical Chemistry, Vol. 25, No. 10, 2006.

Although, there is a growing number of reported applications of SSNMR in research, about 50-100 articles per year in the last five years, but it is still not a readily available method all over the world. So, it cannot be expected to be an essential requirement of its use for the characterization of solid samples.

Reviewer 3. made a very general and rather disparaging statement about the interpretation of the vibrational spectral changes in our samples. It would be true if the peaks in question would have been negligible of importance, e.g. belonging to the deformation modes of the methyl-groups present in the drug. Reviewer 3. is in error, when she or he claims that the peaks originating from the out of plane deformation modes of the aromatic rings of the drug are of negligible importance! The only way these modes can be altered, is that the main body of the drug, those rings, were affected by the changes. She or he also neglected the fact, that the intensity changes of those bands follow an autocatalytic kinetics, characteristic on phase transitions in solid state! Proper references and explanation are given on page 10, in the manuscript!

So, a new solid phase has formed, which affected the aromatic rings of the drug, which is only possible if the drug entered the internal, non-polar cage of the CD-s! It was also clearly deduced in the manuscript.

Reviewer 3. also stated that these spectral changes could be interpreted by amorphization. There are two problems with it. First, the manual grinding method, using mortar and pestle is less efficient in transferring energy than other, mechanical mills. See page 3 paragraph 1 of Prof. Mura’s article quoted by Reviewer 3. Amorphization was not experienced either during the preliminary grinding of TER, prior preparing the physical mixtures of the ingredients. See spectra of pure crystalline TER and the samples marked by 0 min., in Figure 5. on page 9, in the manuscript.

Reviewer 4 Report

The paper describes the development of a solvent free method to produce cyclodextrin complexes of a low water solubility drug, terbinafine, in order to improve its biopharmaceutic properties. The authors have prepared the cyclodextrin complexes followed by their exhaustive characterization in terms of physicochemical behavior. The analytical tools used in their assessment are all consecrated methods in the field of cyclodextrin complexes thus providing reliable information on the final compounds. The authors have designed a scientifically sound chain of experiments which collectively validate the formation of inclusion complexes with optimized dissolution and in vitro diffusion and thus an improved biopharmaceutic profile. The experiments are clearly presented in a logical succession, providing reliable and reproducible results which were later discussed against the previously published literature data. The conclusions are fully supported by the results. I would recommend however the checking of the entire text in terms of English language topic and grammar in order to increase the readability and fluence of the paper.

Author Response

Reply to Referee comments

Reviewer 4

Thank you for taking the time and effort to carefully check and review our manuscript. We also appreciate your positive overall opinion of our work. Below are the answers for your questions whereas modifications on the manuscript are highlighted by blue colour.

I would recommend however the checking of the entire text in terms of English language topic and grammar in order to increase the readability and fluence of the paper.

            Thank you for your remarks, we made changes to some parts of the manuscript that resulted in English correctness and readability.

Round 2

Reviewer 3 Report

Overall, the reply to the comments made to the original manuscript is satisfactory. I agree that the system under study is challenging. Data regarding dissolution and permeability studies of the physical mixtures should be added to the main body or at least to the supplementary information just to be fair in the comparative analysis. I still believe that conclusions and the title of the article is an overinterpretation of the findings “of a new solid phase”. A large number of CD inclusion complexes have been already characterized in the literature, several of them with fine structural details so it is not an impossible task. It is just a matter of scientific rigor. If SS-NMR studies are not possible, a computer modelling of the complex would give a better idea of the feasibility of formation of the inclusion complex or alternative complexation modes with CDs. Note that József Szejtli in his landmark review article (Pure Appl. Chem., Vol. 76, No. 10, pp. 1825–1845, 2004) already warned about the phenomena occurring with “Ciclodextrin Literature” (p. 1838) so advances in the field will not come just by piling up redundant information but from proper characterizations leading to a better understanding of new phenomena.

Author Response

Reply to Referee comments

Reviewer 3

Thank you for taking the time and effort to carefully check the answers and the revised version of the manuscript. Below are the answers for your questions whereas modifications on the manuscript are highlighted by yellow colour.

Overall, the reply to the comments made to the original manuscript is satisfactory. I agree that the system under study is challenging. Data regarding dissolution and permeability studies of the physical mixtures should be added to the main body or at least to the supplementary information just to be fair in the comparative analysis.

Thank you for your remarks. Data regarding in vitro studies have been added to supplementary information.

I still believe that conclusions and the title of the article is an overinterpretation of the findings “of a new solid phase”.

Based on these comments the title of the manuscript has been changed to “Development of Solvent-free Co-ground Method to Produce Terbinafine Hydrochloride Containing Cyclodextrin Binary System; Structural and In Vitro Characterizations”.

A large number of CD inclusion complexes have been already characterized in the literature, several of them with fine structural details so it is not an impossible task. It is just a matter of scientific rigor. If SS-NMR studies are not possible, a computer modelling of the complex would give a better idea of the feasibility of formation of the inclusion complex or alternative complexation modes with CDs. Note that József Szejtli in his landmark review article (Pure Appl. Chem., Vol. 76, No. 10, pp. 1825–1845, 2004) already warned about the phenomena occurring with “Ciclodextrin Literature” (p. 1838) so advances in the field will not come just by piling up redundant information but from proper characterizations leading to a better understanding of new phenomena.

Thank you for your insightful remarks. Structural analysis of cyclodextrin inclusion complexes by computer modelling is indeed a promising field and is encountered in more and more articles (https://doi.org/10.1016/j.carbpol.2021.118644). In the future, we would like to further refine our manuscripts with similar studies, for which we are looking for cooperation partners.
